# Psychological Distress Related to the COVID-19 Pandemic: The Protective Role of Hope

Luca Flesia [1,*], Muhammad Adeeb [2], Aqsa Waseem [2], Mai Helmy [3,4] and Merylin Monaro [5]

1   Azienda ULSS6 Euganea, Via Degli Scrovegni, 14, 35131 Padova, Italy
2   School of Applied Psychology, Social Work and Policy, University of Utara, Sintok 06010, Malaysia
3   Psychology Department, Faculty of Arts, Menoufia University, Menoufia 32511, Egypt
4   Psychology Department, College of Education, Sultan Qaboos University, Muscat P.O. Box 50, Oman
5   Department of General Psychology, University of Padova, 35131 Padova, Italy
*   Correspondence: luca.flesia@ordinepsicologiveneto.it

**Abstract:** The COVID-19 outbreak and the worldwide lockdown measures had an impact on the global mental health and psychological well-being of the general population. Several studies attempted to investigate the protective and risk factors for psychological distress related to the pandemic. However, to date, little is known about the role of hope in this context. The aim of this study was to determine the relationship between hope and psychological distress related to the COVID-19 outbreak in the general population. The sample consisted of 504 Pakistani people who completed cross-sectionally the COVID-19 Peritraumatic Distress Index (CPDI) and the Adult Hope Scale (AHS). Bivariate Pearson correlation analysis was run to measure the relationship between hope and psychological distress; hierarchical regression analysis was run to investigate the association between demographics and hope with psychological distress. Higher levels of hope predicted lower levels of psychological distress. Being female, being older, lower level of education, urban residence, being married and living in nuclear family systems were associated with higher levels of psychological distress. The study highlights the protective role of hope on psychological distress related to COVID-19, contributing to knowledge on factors promoting positive mental health during emergency times and providing useful information for implementing effective public health policies and programmes.

**Keywords:** psychological distress; hope; COVID-19; pandemic; mental health; well-being

## 1. Introduction

The COVID-19 pandemic was an extraordinary event, arising unexpectedly and creating a worldwide outstanding public health concern. Strict public health measures were taken to contain the global spread of the SARS-CoV-2 virus, which was declared a pandemic by the WHO on 11 March 2020 [1]. The constantly increasing number of cases and deaths has led authorities to impose social distancing and home confinement [2]. Alongside concerns about financial security, people had their daily routines disrupted indefinitely and were abruptly isolated from people and places that were part of their daily lives [3]. All of these changes seriously affected people's mental health, especially with regard to anxiety-related symptoms [4]. Given this, investigating the resilience and protective factors of people's mental health during the COVID-19 outbreak represents a relevant and impactful issue. Consistent with this, the goal of this paper was to investigate the role of hope in influencing the psychological distress related to the COVID-19 pandemic. Hope is a dynamic mental process, indicating the individual perception that own personal efforts will lead to positive futures (i.e., pathways and agentic thinking) [5]; it is the sum of three integrated elements: goals, agency and routes (strategies, visions, plans) [6]. Psychological distress is defined as a state of emotional suffering characterised by symptoms of depression (i.e., sadness, hopelessness, lost interest) and anxiety (i.e., feeling tense, restlessness), often

associated with somatic symptoms (i.e., headaches, insomnia), consequent to exposure to a stressful event that threatens physical or mental health [7,8]. Psychological distress related to COVID-19 specifically refers to behaviours, emotions, thoughts and symptoms associated with stress linked to the COVID-19 outbreak [9].

Pandemics are reported to be extremely stressful events that force people to cope with totally unexpected, ambiguous and uncertain situations [10]. Specifically, the literature underlines two main aspects of the COVID pandemic's impact on people's mental states. The first is related to danger (i.e., the fear of contagion), which can increase perceived threat and sometimes lead to panic, behavioural contagion and emotional epidemic [11,12]. The second regards the multiple and rapid changes to social, working and familiar habits due to self-isolation and social distancing measures [13,14]. The longer the duration of self-isolation, the more people are likely to experience frustration and boredom, along with concerns about infection [15]. Well-documented psychological reactions to epidemics include emotional distress, anxiety behaviours, sleeping disorders, fear, anger, depression, health concerns, a sense of powerlessness and uncertainty [14,16–18]. Moreover, studies examining the long-term consequences of infectious epidemics showed that some individuals might even develop symptoms of post-traumatic stress disorder (PTSD) [19–21] lasting up to 3 years following the end of the epidemic [15–18].

Several studies have attempted to investigate protective and risk factors for psychological distress related to the COVID-19 pandemic, investigating the role of personality and other stable psychological traits, such as coping styles, in influencing the impact of the COVID-19 pandemic on people's mental health [22–26]. Attitude towards the future can also be a relevant factor in influencing people's reactions to critical events and situations and, therefore, in influencing psychological distress. According to Snyder's cognitive model of hope, hope refers to "a motivational positive state that is based on an inter-actively derived sense of successful (i) agency (goal-oriented energy) and (ii) pathways (preparation to achieve goals)" [5]. Hope has been increasingly recognised as an important positive factor in promoting well-being and psychological adjustment, as well as an important resource in coping with stress and uncertainties [27,28]. Past studies have revealed that hope (measured through the Adult Hope Scale) had a significant positive relationship with subjective/psychological well-being and a negative relationship with psychological distress [29–33]. Plenty of past research has indicated that lower amounts of hope—or the presence of hopelessness—are positively connected with a raised risk of mental well-being issues, such as anxiety, depression and post-traumatic stress disorder [34,35]. Research has shown that hope could contribute to enhanced physical as well as mental well-being and help in adjustment when dealing with stressors [36,37]. Moreover, hope has been identified as a protective factor against psychological distress associated with negative life events, natural disasters and psychiatric disorders [35,38–40].

Despite the significant influence of hope in promoting well-being and psychological adjustment and the significant uncertainty and unpredictability towards the future that characterise pandemics and lockdowns, very little research has investigated the role of hope in the COVID-19 scenario. To investigate hope levels following the unexpected changes in daily lives related to the lockdown period, Amirav et al. [41] compared depression and hope levels before and during the COVID-19 pandemic in a large sample of European adults: contrary to expectations, the authors found that despite increased depression, the COVID-19 lockdown was associated with significantly high hope levels. Mirhosseini et al. [42] analysed cross-sectionally a large sample of Iranian adults, finding a direct association between high levels of hope and low anxiety scores. Nearchou and Douglas [43] investigated the role of hope in buffering the relationship between the traumatic distress of COVID-19 and depression in the general Irish population, finding a protective role of hope. Laslo-Roth et al. [44] also found a protective effect of hope on psychological distress among a sample of Lebanese adults. Although data from the literature seem to suggest a protective role of hope on psychological distress related to the COVID-19 pandemic, more data are needed to confirm this relationship. Moreover, the role of demographic

variables in influencing this association needs to be investigated. Finally, to the best of our knowledge, to date, there are no data available in regard to the relationship between hope and psychological distress in the Pakistani population. As Pakistan is a developing country characterised by specific economical and societal features, especially in comparison with Western countries [45,46], this aspect could be worth investigating. Such investigation will enrich the existing literature, enlarging our comprehension of mental health's protective factors to the impact of the COVID-19 outbreak. Moreover, a better understanding of the association between hope and psychological distress related to the COVID-19 pandemic could provide significant information to define effective preventive campaigns, improving people's well-being and reducing public health costs.

To fill this gap, the main aim of the current research was to investigate the relationship between hope and psychological distress related to the COVID-19 outbreak in the Pakistani adult population. Specifically, we hypothesised that:

**Hypothesis 1 (H1).** *Higher levels of hope are associated with lower levels of psychological distress related to the COVID-19 outbreak.*

A large amount of literature also investigated the role of demographic features in predicting psychological distress related to the COVID-19 pandemic, documenting some categories of individuals as being more vulnerable than others. Specifically, female people, younger people, people with low education, people from urban residences, people living with others and people who are not engaged in a relationship were reported to manifest higher levels of mental distress than their counterparts [22,47–49]. Moreover, healthcare workers are reported to be more exposed to risks of mental health distress [50]. The second aim of this research was to investigate the role of demographic variables in predicting psychological distress related to the COVID-19 pandemic. In Pakistani society, the family forms the foundation of society and encompasses a wide breadth of relationships. One's extended relatives have great significance on a daily basis: indeed, a vast majority of Pakistanis live in multigenerational households whereby three, four or sometimes five generations reside together (including grandparents, uncles, siblings and cousins). Specifically, the term "joint family" refers to this form of family in which members of a unilineal descent group (typically the male one) live together with their spouses and offspring in one homestead and under the authority of one of the members. Compared to the international literature, this sociocultural specificity could influence the association of cohabitation status and marital status with psychological distress related to the COVID-19 outbreak. As regards gender, age, level of education and residence, on the contrary, our hypothesis is consistent with the literature. Therefore, we hypothesised that:

**Hypothesis 2 (H2).** *Females, younger participants, participants with low education and participants in urban residences will report higher levels of psychological distress than their counterparts.*

## 2. Materials and Methods

### 2.1. Participants

A total of 557 respondents accessed the survey; 53 subjects were excluded for the following reasons: being under 18 years of age (n = 22), not having completed the questionnaire fully (n = 27) or having withdrawn their consent (n = 5). The final analytic sample consisted of 504 Pakistani participants (average age = 30.56, SD = 6.30).

Participants provided information on their sex assigned at birth (male, female), age (in years), education (intermediate, graduation, master), residence (urban or rural), family system (nuclear family or joint family) and marital status (single, married, divorced). As regards residence, in line with Shafqat et al. [51], participants were informed that "urban residence" referred to people living in city areas (e.g., Lahore, Faisalabad, Islamabad, Karachi, Bahawalpur and Multan), while "rural residence" referred to people living in village areas [51]. As regards family system, in line with Wasim et al. [52], respondents were

informed that "joint family" referred to people living with parents, grandparents, uncles and cousins, while "separate family" referred to people living only with nuclear relatives (e.g., parents and siblings). Detailed results of the frequency distribution of demographic variables are reported in Table 1.

*2.2. Instruments*

2.2.1. Psychological Distress

The psychological distress related to the COVID-19 pandemic was measured by using the COVID-19 Peritraumatic Distress Index (CPDI) by Qiu et al. [4]. The questionnaire consists of 24 items inquiring about anxiety, depression, specific phobias, cognitive change, avoidance and compulsive behaviour, physical symptoms and loss of social functioning, referring to the last week (example item: "compared to usual, I feel more nervous and anxious"). Responses are given on a 5-point Likert scale ranging from 0 (not at all) to 4 (extremely). The questionnaire scores range from 0 to 96. Scores between 28 and 51 indicate mild to moderate distress; scores ≥52 indicate severe distress. The Cronbach's alpha of CPDI is 0.95 [4].

2.2.2. Hope

Participants' level of hope was measured using the Pakistani version of the Adult Hope Scale (AHS) by Snyder et al. [5]. It is based on Snyder's cognitive model of hope. The scale consists of 12 items: four items measure pathways thinking, four items measure agency thinking and four items are fillers. Participants respond to each item using an 8-point Likert scale ranging from "definitely false" to "definitely true" (example item: "I can think of many ways to get out of a jam"). The questionnaire scores range from 8 to 64. Total scores between 40 and 48 indicate hopeful state, 48–56 moderately hopeful state, 56 or higher high hope state; scores below 40 indicate low hope. Snyder et al. reported the Cronbach alphas for the total score ranged from 0.74 to 0.84 [5]. AHS was also translated into Urdu language (national language of Pakistan) and validated among Pakistani population: the reliability of the Pakistani version of AHS was found acceptable in pathway (alpha = 0.88), agency (alpha = 0.85) and overall hope (alpha = 0.86) [53].

*2.3. Procedure*

Data collection took place cross-sectionally using an online survey managed through Google Forms between 3 March 2020 and 28 April 2020. The survey link was disseminated through social media (i.e., WhatsApp, Facebook) using a snowball sampling technique. Participation was voluntary. Eligible participants were people from the general population aged 18 years or more. It was calculated that a sample size of 160 is minimum to achieve a statistical power $(1-\beta) = 0.95$ in a hierarchical regression analysis involving 7 predictors, given a significance level $\alpha = 0.05$ and a medium effect size (0.15) [54]. Respondents who reported a history of mental illness and/or could not complete the online survey independently were excluded. All participants were asked to read and sign an online informed consent prior to data collection. They did not receive any compensation for participation. According to the ethical guidelines provided by the American Psychological Association, the experimental procedure did not involve any possible harm to individuals, society or study participants. The study received formal approval from the ethical committee of Riphah International University Faisalabad (Riphah-Fsd/Off-2020/1062).

**Table 1.** Frequency distribution of demographic variables.

| Respondent's Characteristics | | *f* (%) |
|---|---|---|
| Gender | Male | 208 (41.3) |
| | Female | 296 (58.7) |
| Education | Intermediate | 73 (14.5) |
| | Graduation | 207 (41.1) |
| | Master | 224 (44.4) |
| Residence | Urban | 285 (56.5) |
| | Rural | 219 (43.5) |
| Family system | Nuclear | 172 (34.1) |
| | Joint | 332 (65.9) |
| Marital status | Single | 390 (77.4) |
| | Married | 104 (20.6) |
| | Divorced | 10 (02.0) |

*2.4. Data Analysis*

The collected data were analysed using SPSS (24.0 version). Before analysing data, the normality assumption was checked using skewness and kurtosis criteria. If the kurtosis is close to 0, then a normal distribution can be assumed [55]. Likewise, normal distributions have skewness close to 0 [55]. Considering these criteria for all variables in the current research, both the values of kurtosis ($-50$ to 1.85) and skewness ($-1.41$ to $-58$) confirmed an adequate range, which supports a normal distribution; therefore, parametric tests were applied. A bivariate Pearson correlation analysis was run to measure the relationship between hope and psychological distress. A hierarchical regression analysis was conducted to measure the predicting role of hope and demographic variables (e.g., age, gender, education, residence, family system and marital status) on psychological distress. Before running the hierarchical regression analysis for moderation analysis considering Baron and Kenny's guidelines [56], the distribution of each demographic variable was checked according to hope. Then, to verify the confounding role of any significant difference, all demographic variables were included in regression analysis.

**3. Results**

*3.1. Descriptive Statistics*

On average, participants reported a mild to moderate level of psychological distress (M = 29.20, SD = 16.50). While 54% of the respondents experienced normal psychological distress, 34.9% of the respondents experienced mild psychological distress and 11.1% of the respondents experienced severe psychological distress. As regards hope, on average, respondents were found in the hope state (M = 41.09, SD = 10.54). While 34.2% of the respondents reported low hope, 32.3% reported hope, 29.2% reported moderate hope and only 4.3% reported experiencing high hope.

*3.2. Association between Hope and Psychological Distress*

To investigate the relationship between hope and psychological distress, a bivariate Pearson correlation analysis was run. Results (Table 2) revealed a significant negative correlation between psychological distress and hope ($r = -0.37$, $p < 0.001$). Moreover, analysing the AHS subscales separately, a significant negative correlation emerged between psychological distress and both agency ($r = -0.30$, $p < 0.001$) and pathways ($r = -0.25$, $p < 0.001$).

**Table 2.** Correlation between psychological distress and hope (N = 504).

| Variables | Psychological Distress | Hope | Agency | Pathways |
|---|---|---|---|---|
| Psychological distress | - | −0.37 ** | −0.30 ** | −0.25 ** |
| M (SD) | 29.20 (16.50) | 41.09 (10.54) | 20.65 (5.98) | 20.45 (5.58) |
| Cronbach's alpha | 0.90 | 0.84 | 0.75 | 0.71 |

Note. ** = $p < 0.001$.

### 3.3. The Association between Demographics and Psychological Distress

To investigate the association between demographics and psychological distress, the distribution of each demographic variable was checked according to hope. The results of the comparison highlighted significant differences in the distribution of hope according to age, gender, education, residence, marital status and family system. Results are reported in the Supplementary Material. The mean scores of hope were significantly higher among males compared to females (hope: $t = 2.33$, $p < 0.05$), significantly higher among young adults compared with middle-aged adults (hope: $t = 3.55$, $p < 0.01$), significantly higher among master's level of education compared with intermediate and graduation level of education (overall: $F = 20.09$, $p < 0.01$), significantly higher among rural residence compared with urban residence respondents (hope: $t = 3.27$, $p < 0.01$), significantly higher among respondents from nuclear family compared with respondents from joint family (hope: $t = 3.64$, $p < 0.01$) and significantly higher among single/unmarried compared with married and divorced respondents (hope: $F = 7.79$, $p < 0.02$).

These variables were then entered as predictors of psychological distress in the hierarchical regression model.

A hierarchical regression analysis was run with psychological distress as the dependent variable and the following independent variables: hope + demographic variables (age, gender, education, residence, family system and marital status). Dummy variables were created for education and marital status because both these demographic variables had more than two groups. Results are reported in Table 3. In step 1, age, gender, education (intermediate and graduation as dummy variables), residence, family system and marital status (married and divorced as dummy variables) were taken from the demographics variables, while in step 2, hope was taken as a further predictor and in step 3, the interaction between demographic variables and hope for moderation was taken. Significant results are reported in Table 3. In step 1, age (B = 0.27, β = 0.15, $p < 0.01$), female gender (B = 1.57, β = −0.08, $p < 0.05$), intermediate education (B = 4.41, β = 0.17, $p < 0.01$) and married as marital status (B = 3.47, β = 0.15, $p < 0.01$) were found significant positive predictors of psychological distress, while rural residence (B = −10.15, β = −0.54, $p < 0.01$) and joint family system (B = −4.44, β = −0.22, $p < 0.01$) were found significant ($R^2 = 0.27$, F (8, 495) 25.69, $p < 0.01$) negative predictors of psychological distress. In step 2, hope (B = −0.20, β = −0.21, $p < 0.01$) was also found to be a significant ($R^2 = 0.30$, F (9, 494) 26.55, $p < 0.01$) negative predictor of psychological distress. Conversely, in step 3, the interaction between hope and demographic variables was tested; the results showed that joint family system (B = −4.06, β = −0.21, $p < 0.05$) was a significant ($R^2 = 0.32$, F (17, 486) 13.71, $p < 0.01$) moderator between hope and psychological distress.

**Table 3.** Hierarchical regression analysis for psychological distress with predicting role of age, gender, education, residence, family system, marital status and hope (N = 504).

| Variables | Psychological Distress | | | | | |
|---|---|---|---|---|---|---|
| | $R^2$ | B | β | F | *p* | 95% CI |
| Step 1 | 0.27 | | | 25.69 | 0.00 | |
| Age | | 0.27 | 0.15 | | 0.00 | (0.13, 0.41) |
| Gender (female) | | 1.57 | 0.08 | | 0.04 | (0.11, 3.03) |
| Education (intermediate) | | 4.41 | 0.17 | | 0.00 | (2.37, 6.45) |
| Residence (rural) | | −10.15 | −0.54 | | 0.00 | (−12.13, −8.18) |
| Family system (joint family) | | −4.44 | −0.22 | | 0.00 | (−6.49, −2.37) |
| Marital status (married) | | 3.47 | 0.15 | | 0.00 | (1.69, 5.27) |
| Step 2 | 0.30 | | | 26.55 | 0.00 | |
| Hope | | −0.20 | −0.21 | | 0.00 | (−0.28, −0.12) |
| Step 3 | 0.32 | | | 13.71 | | |
| Hope _X_ joint family system | | −4.06 | −0.21 | | 0.04 | (−8.14, −0.02) |

Age (continuous variable); gender (male = 0, female = 1); education as dummy variable (bachelor = 0, master = 0, intermediate = 1); residence (urban = 0, rural = 1); family system (nuclear family system = 0, joint family system = 1); marital status as dummy variable (single = 0, divorced = 0, married = 1).

Figure 1 represents the graphical explanation of moderation; it shows that the relationship between hope and psychological distress is influenced by the moderation of the family system. It indicates that hope contributes to reducing the levels of psychological distress more for people living in joint families than for people living in nuclear families.

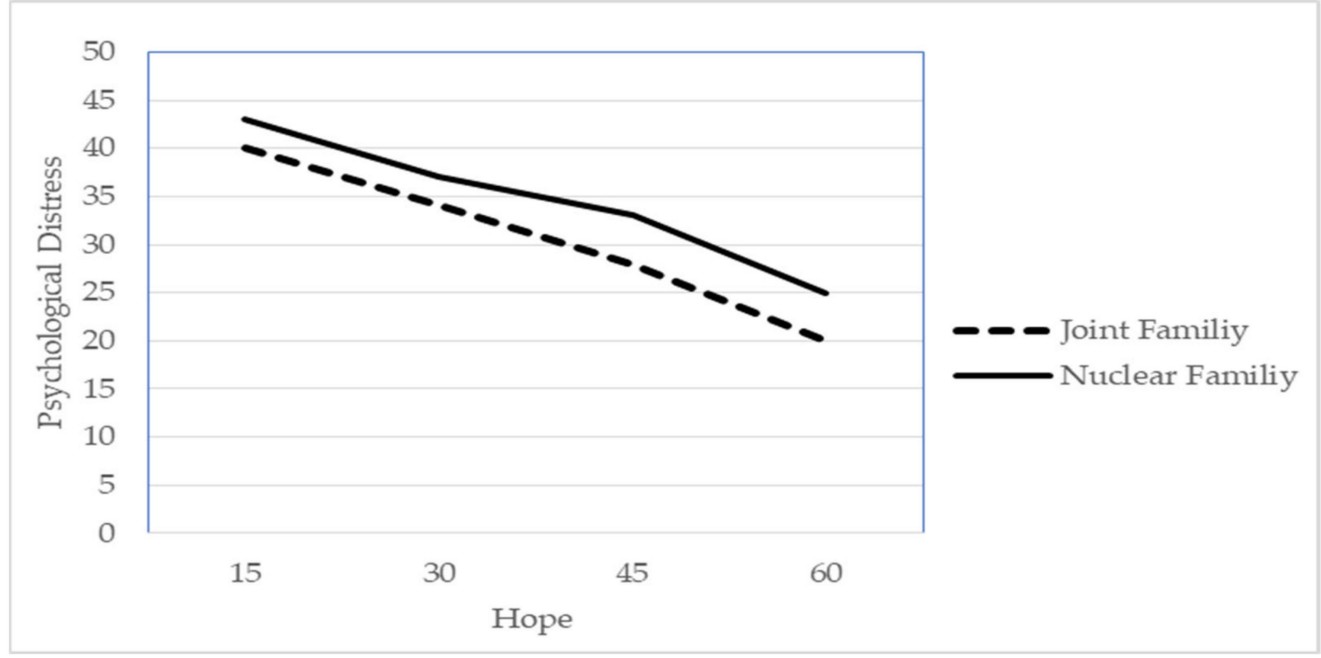

**Figure 1.** Interaction between hope and joint family.

## 4. Discussion

The aim of this study was to investigate the relationship between hope and psychological distress related to the COVID-19 pandemic in a sample of the adult Pakistani population. Overall, the study revealed a mild level of psychological distress within the sample: while 54% of respondents declared to experience normal levels of psychological distress, 34.9% of respondents referred to moderate psychological distress and 11.1% to

severe levels of psychological distress. As compared with studies coming from different countries, we found higher levels of distress. For example, Costantini and Mazzotti [9], investigating the level of psychological distress during the COVID-19 pandemic in a sample of Italian adults, reported that participants experiencing mild/moderate or severe levels of psychological distress (as measured with CDPI) were about one-third of the sample. A similar countrywide study conducted in China using CPDI confirmed lower levels of psychological distress, with 35% of participants declaring moderate to severe levels of psychological distress [4]. Of note, according to Marzo et al. [49], levels of psychological distress can significantly differ from one country to another; the authors conducted a study using CPDI among the general population of 13 countries, finding rates ranging from 94.5% (Vietnam) to 14.1% (Nepal). Other studies investigating psychological distress within the Pakistani population also found high levels of psychological distress [57]. Previous studies documented that personal ability to satisfy one's own basic needs (i.e., financial security and physical safety) is a significant predictor of psychological distress [15,58]. The economic poverty the Pakistani population falls into might explain our results. Consistent with this explanation, Mamun [46] found that lockdown-related economic recession and subsequent distress were the first causes of suicides during the first wave of the COVID-19 pandemic in Pakistan.

Other sociocultural differences, such as social connectedness and family organisation, might buffer the impact of the COVID-19 pandemic on people's psychological distress and, therefore, contribute to explaining countrywide differences. For instance, social connectedness has been recognised as a significant protective factor towards psychological distress during the COVID-19 pandemic, especially in contrast to social isolation and the impact of lockdown measures [59,60]. In this regard, the prevalent family structure of the Pakistani population is the joint family, which is an extended structure where many people, including three or more generations (grandparents, sons with their wives and children and unmarried siblings) reside together [61], can provide a sense of belongingness and connectedness [60]. Consistent with this, we found that people living in nuclear families reported higher levels of psychological distress than their counterparts. Moreover, we found an interaction effect based on the family system: hope predicted lower levels of psychological distress more effectively for people living in joint families than for those living in nuclear ones. This result is consistent with research by Shakil et al. [62], finding severe levels of psychological distress among Pakistani people living in nuclear family systems. This result suggests the need to target specific interventions for promoting hope, especially for people living in nuclear families. Future studies should better investigate the role of sociocultural variables as well as their interplay in influencing people's COVID-related psychological distress.

Consistent with the previous literature, as well as with H2, being female, having a lower level of education and living in a rural area were associated with higher levels of psychological distress. Several studies reported that women tended to experience higher levels of psychological distress than men during the COVID-19 pandemic [4,14,63–66], as well as in nonpandemic situations [67–69]. Past studies reported that low education is positively related to psychological distress [70,71]. Studies conducted during the COVID-19 pandemic have also shown that respondents with low education are more likely to have higher levels of psychological distress [14,49,72]. In terms of human capital, education promotes well-being through skills, resources and good habits that permit persons to improve their effectiveness [73,74]. Moreover, in Pakistan, having a higher level of education tends to be associated with higher family socioeconomic status [75]. This could be an underlying protective factor in the association between level of education and psychological distress, buffering people's economic worries subsequent to the pandemic. As highlighted in the literature, people living in rural areas tend to have more greenspaces and wider living areas, which is a protective factor during lockdown times [76,77]. Moreover, people living in urban areas might be more exposed to crowded and social situations (i.e., supermarkets, buses) and have a higher perceived risk of contracting the virus [48]. Indeed, higher

population mass in urban places is known to facilitate the spread of viruses [78]. Of note, considering the Pakistani population, people living in joint families are more likely to live in rural areas as well [79].

As regards age, H2 is rejected. Older participants reported higher levels of psychological distress than younger ones. These results are in contrast with previous studies indicating higher levels of psychological distress among younger people [22] and with research suggesting that older persons have less chance of stressful life occasions, use more coping strategies and take significant advantage of life experiences and experience of public crises. All of this would help them to ease psychological distress during the COVID-19 pandemic [80,81]. However, the results of the present study are in line with those of Twenge and Joiner, highlighting higher levels of mental distress among middle-aged adults (30–44 years) compared with young adults (18–29 years) [68]. Past studies also reported more psychological distress for older people compared with younger people [82]. Kluge [83] claimed that the risk factor of death and the high rate of illness from COVID-19 rise with age, resulting in psychological distress among middle and older age adults. Past studies support the results of the present research that age has a positive relationship with psychological distress [4,84,85]. Of note, our overall sample consisted of globally middle-aged people; this issue might have impacted the study results.

As regards hope, 34.2% of participants reported low levels of hope. These data might indicate the negative impact of the pandemic on people's hope. However, as compared with data coming from other countries, the results of the current study showed lower levels of hope in the general Pakistani population. For example, Yildirim and Arslan [86], investigating the levels of hope among the Turkish adult population during the COVID-19 outbreak, found globally moderate levels of hope, as measured with the AHS considering the same scoring criteria as the present research. As well as for results on the levels of psychological distress, this result could be explained based on the widespread poverty that overall characterises the Pakistani population. Future studies should investigate the possible underlying and moderating factors of such countrywide differences, for instance, focusing on the role of economic status and its interplay with other sociocultural factors (e.g., religion, health services).

Concerning the association between hope and psychological distress, our results support hypothesis 1: higher levels of hope were associated with higher levels of psychological distress related to the COVID-19 pandemic. During the early stages of the COVID-19 pandemic, people were exposed to many unprecedented stressful events: the virus was unknown, contagious and dangerous; fear of contagion; social isolation; and insecurity towards the future significantly impacted people's everyday lives and scenarios. All these aspects are reported to have affected people's beliefs and expectations for the future [87]. Our results align with data coming from previous studies that investigated the association between hope and psychological distress [42–44,88,89]. Therefore, the present study contributes to the little existing literature on the topic, corroborating the theory about a negative association between hope and COVID-related psychological distress. Hope is a source of strength: when we face life events with hope, we face them with a sense that there is something we can do about them; this feeling makes life "easier to live [89,90]". Future studies are needed to better investigate factors influencing the levels of people's hope, both in general and specifically in the COVID-19 scenario. Such efforts could help to efficiently promote people's mental health and well-being, even in adverse times. The present study highlights the relevance of personal attitudes towards stressful events, suggesting the need to implement preventive campaigns to increase people's awareness of the relationship between triggering events, personal beliefs and consequences. Such campaigns could help people understand that underlying beliefs affect how they think about and respond to events and could, therefore, increase people's resilience towards stressful events, both in pandemic and nonpandemic situations.

This study has several limitations. First, participants were recruited via social media tools. Consequently, self-selection bias may have to be considered, affecting the repre-

sentativeness and generalisability of the results. Second, the population sample is small compared with the sample community; this also limits the representativeness and generalisability of results. Third, there were few demographic differences in regard to age, as participants were aged between 18 and 55; thus, regarding age, the sample was largely homogenous. Despite this, we found a different pattern in stress responses to the pandemic within this relatively homogenous group, which is an interesting result. Finally, the study was based on cross-sectional research; therefore, it is not possible to detect causal relationships among variables.

## 5. Conclusions

The present research contributed to research investigating the impact of the COVID-19 pandemic on people's psychological distress and related influencing factors. The study results highlighted that psychological distress was at a mild level among Pakistani adults during the pandemic situation of COVID-19 and inversely associated with hope. Female adults, individuals from urban residences, married people, people with lower education and people living in nuclear families were more prevalent in psychological distress as compared with their counterparts. Given the significant role of hope in protecting against psychological distress in pandemic situations, this research may be helpful for decisionmakers to effectively prepare preventive programmes and overcome COVID-19 negative mental outcomes by enhancing the resilience of common emotions and by urgently training health professionals to provide adequate care bases for risk groups and affected individuals. Future studies should be focused on interventions based on the domain of positive psychology to reduce psychological distress. The COVID-19 pandemic has had a long-term effect on people's mental health; therefore, psychologists and health professionals must conduct seminars to boost hope among the general population.

**Supplementary Materials:** The following supporting information can be downloaded at: https://www.mdpi.com/article/10.3390/ejihpe13010005/s1, Table S1: title; Distribution of Demographics according to Hope and Psychological Distress.

**Author Contributions:** Conceptualisation, L.F. and M.A.; methodology, L.F. and M.A.; formal analysis, M.A.; investigation, L.F., M.A., A.W., M.H. and M.M.; writing—original draft preparation, L.F. and M.A.; writing—review and editing, L.F., M.A. and M.M.; supervision, L.F.; project administration, L.F. All authors have read and agreed to the published version of the manuscript.

**Funding:** This research received no external funding.

**Institutional Review Board Statement:** The study was conducted in accordance with the Declaration of Helsinki and approved by the ethics committee of Riphah International University Faisalabad (protocol code Riphah-Fsd/Off-2020/1062; date of approval 5 May 2020).

**Informed Consent Statement:** Informed consent was obtained from all subjects involved in the study.

**Data Availability Statement:** Data will be made available upon reasonable request to the corresponding author.

**Conflicts of Interest:** The authors declare no conflict of interest.

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
