# Peer review of "Psychological Distress Related to the COVID-19 Pandemic: The Protective Role of Hope"

_ejihpe, doi:10.3390/ejihpe13010005_

Round 1
Reviewer 1 Report
· In general, there are some phrases that don’t read as smoothly and could use some revision to ensure appropriate English syntax. Example: Sentence beginning on Line 60, the word resulted makes the sentence somewhat awkward.
· Throughout the introduction, it would be helpful if the authors made it clear when the literature to which they are referring are studies that were specifically published surrounding COVID findings.
· The introduction would benefit from the very first paragraph making it clear what the goal of the study is and what constructs will be discussed. As written, it is a little unclear what different topics will be covered and how they all relate. I agree that these constructs are all important but without theoretical frameworks and goals stated in the beginning it feels a little less cohesive.
· In line 66, there is discussion of losing one’s optimistic side. Hope and optimism are two distinct concepts. If the authors include this citation, they would also need to make it clear how hope and optimism are different constructs, and why they are focusing on hope.
· On line 102, there is a statement about the relation not being examined in Pakistani population. This needs more information/support as to why this might look different (or the same). As it is worded, it almost sounds like the authors believe the patterns wouldn’t be similar to studies from other populations.
· It would be helpful early in the manuscript to explicitly say the operational definition of psychological distress so that when the reader is reading the hypotheses, they know what constructs are included.
· H2 seems very lumped together and as though it was just taken from the intro. In the introduction there should be more rational as to why these demographic variables/characteristics were chosen and lumped together.
· I am unsure what a joint family system is.
· The last paragraph of the introduction may be better situated earlier in the introduction to highlight the purpose of the study.
· There is no mention of the number of participants who completed the survey, who were included in the analytic sample, or who clicked the link but didn’t complete the survey.
· Line 150: The authors should report the alpha for their sample.
· Line 160: The authors should report the alpha for the hope scale.
· In reading the measures section it is clear that Psychological Distress is specifically Psychological Distress related to COVID-19. This isn’t clear in the introduction, and I was expecting general measures of anxiety and depression given the literature review. This needs to be clarified in the introduction and likely the introduction slightly rewritten to make it explicitly clear that these are specific to COVID and the topics that are included in it.
· The AHS was created for a US population, and I am unsure about the CPDI. The authors should include information regarding the validity and reliability of these measures for a Pakistani population. The CPDI might be harder, given that it was time sensitive, but it is critical that the AHS has been shown to be valid for this population.
· Line 179: There is no mention of normality of the scales. The statistics used required normality assumptions and often hope and distressed tend to be skewed, which would mean that Pearson correlations and regressions are not appropriate for the data.
· Line 197: I’m unsure that the sentence that beings with “First” means.
· I’m curious why the authors didn’t also look at mean level demographic difference in psychological distress. This likely looks different for different groups.
· Since all data were collected at a single time point, the researchers should make a much stronger rationale as to why they investigated hope predicting distress vs. distress predicting hope. Or, even stronger, would be for the researchers to investigate both.
· Instead of a hierarchnical linear regression, I would recommend that the researchers utilize moderation or interaction terms to better understand how these demographic variables interact with hope (or with psychological distress). Additionally, if using regression, any variables that have more than 2 groups need to be dummy coded (contrast coded) and have a reference group. It doesn’t appear that this was done and this is a flaw that must be corrected. Additionally, it would be helpful to know what group was the reference group (i.e., 0) for all demographic variables to allow for an individual to interpret what going from one group to the other would mean in terms of the outcome.
· The first paragraph of the discussion doesn’t speculate as to why the levels of psychological distress might look different in this population although the paragraph mentions that. More speculation needed.
· Line 254: The researchers say that other studies found moderate levels of hope. Did those researchers also break up the scale in the same manner as this study? Additionally, there is a lack of speculation surrounding the findings in the paragraph starting on line 249.
· In general, the entire discussion should include more speculations about the findings and the implications for future work. Additionally given the comment about the lack of dummy coding, all discussion that surrounds martial status and education needs to be reexamined after the model is rerun.
· The authors tease apart the results for demographic variable talking only about one at a time, yet they were all included in the same model. This doesn’t allow for interpretation of the full picture of what occurred for those who might be in varying groups.
· The discussion currently reads as a reporting of results with limited speculation/implication discussion.
Reviewer 2 Report
I would like to thank the authors for this research that aims to investigate the relation between hope and psychological distress related to the COVID-19 outbreak in Pakistani adult population.
The research admits that a better understanding of the association between hope and psychological distress related to the COVID-19 pandemic could provide significant information to define effective preventive campaigns, improving people’s wellbeing and reducing public health costs.
The research subject is timely, innovative, and interesting. It also fits the aim and scope of the journal.
The research is well designed and follows a sound scientific research method.
Results and recommendations are clear and could have an impact among the community of researchers.
However, some modifications are needed in order to improve the quality of the paper.
Lines 66, 91, 95, Put references directly after the name of the authors.
Lines 106 – 111, We no longer put hypotheses at the end of the literature review. You need to insert your hypotheses after each related literature review section.
Limitations: You also need to mention that the population sample is small compared to the studied community.
Round 2
Reviewer 1 Report
I appreciate the author(s) responsiveness to the comments/suggestions. I believe that the introduction reads much clearer and rationale is stronger. However, I do believe that the manuscript could benefit from some additional updates.
-I appreciate the inclusion of the dummy coded variables, however they were not coded appropriately. When utilizing dummy coded variables, the number of variables included in the model should be one less than the total number of groups. Further, in dummy coding, there is only one group that is the reference group (0). Given this, the analyses using the dummy codes needs to be rerun appropriately. Additionally, each variable would need its own line in Table 3.
-Per the original review, would like to see the authors utilize a more robust statistical approach to answering their question, such as moderation, to better understand how these demographic variables interact with hope (or with psychological distress). Given the sample size, this should be possible while retaining appropriate statistical power.
-Given the incorrect analyses, I did not review the sections of the manuscript that discuss the model including the improperly created dummy codes, and the results could change and the discussion would need to be modified.
-I appreciate the additions in the discussion. It may be helpful to break up the first paragraph, as its a little long.
-After reading the limitations and the statement around age, I think it would make more sense that the authors utilize age as a continuous variable, especially given that it is truly continuous.
